# Charge transfer from the carotenoid can quench chlorophyll excitation in antenna complexes of plants

Lorenzo Cupellini [1*], Dario Calvani [1], Denis Jacquemin [2] & Benedetta Mennucci [1*]

The photosynthetic apparatus of higher plants can dissipate excess excitation energy during high light exposure, by deactivating excited chlorophylls through a mechanism called non-photochemical quenching (NPQ). However, the precise molecular details of quenching and the mechanism regulating the quenching level are still not completely understood. Focusing on the major light-harvesting complex LHCII of Photosystem II, we show that a charge transfer state involving Lutein can efficiently quench chlorophyll excitation, and reduce the excitation lifetime of LHCII to the levels measured in the deeply quenched LHCII aggregates. Through a combination of molecular dynamics simulations, multiscale quantum chemical calculations, and kinetic modeling, we demonstrate that the quenching level can be finely tuned by the protein, by regulating the energy of the charge transfer state. Our results suggest that a limited conformational rearrangement of the protein scaffold could act as a molecular switch to activate or deactivate the quenching mechanism.

[1] Università di Pisa, Dipartimento di Chimica e Chimica Industriale, Via G. Moruzzi 13, 56124 Pisa, (PI), Italy. [2] Laboratoire CEISAM-UMR CNRS 6230, Université de Nantes, 2 Rue de la Houssiniere, BP-92208, F-44322 Cedex 3, Nantes, France. *email: lorenzo.cupellini@unipi.it; benedetta.mennucci@unipi.it

Nonphotochemical quenching (NPQ) is one of the most intriguing mechanisms of plant photoprotection[1–4]. It encompasses several processes taking place at different timescales after the exposure of the Photosystem II to intense light conditions. Its fastest component, energy-dependent quenching (qE), involves the quenching of the excited chlorophylls before they can convert into their triplet states. The latter could react with oxygen and generate harmful reactive oxygen species.

Although details on the molecular mechanism of qE are constantly being uncovered, there is still debate in the literature about both the nature of the quencher within the antenna complexes, and the exact quenching mechanism[5–8]. It is however clear that a major role is played by the accessory pigments in photosynthesis, the carotenoids. The latter are in fact characterized by a very short-lived excited state (the lowest singlet state or $S_1$), which makes these molecules perfect potential quenchers. Indeed, excitation energy transfer (EET) from the lowest-energy chlorophylls to the neighboring carotenoids has been suggested to be responsible for quenching the chlorophyll excitation in different light-harvesting complexes[9–11]. Another proposed mechanism, which still involves the carotenoids, is instead based on the electron transfer from the carotenoid to the chlorophyll, which is followed by charge recombination in the ground state[12–15].

The uncertainty about the quenching processes and the molecular mechanisms beyond them arise from the difficulty of directly observing and quantifying the single energy or charge-transfer processs. What is instead usually available, in single-molecule experiments, is the lifetime of the whole complex: it is exactly on the measure of this quantity that it has been recently suggested that the antenna complexes can change their main function, from light harvesting to quenching, by changing their conformation[16–20]. This change of function has to be reversible, and nowadays it is widely accepted that it is triggered in vivo by a proton gradient between lumenal and stromal sides of the photosynthetic membrane, possibly mediated by the interaction with the PsbS protein[6,21].

Due to the difficulty in revealing the quenching mechanisms through a direct observation, computational simulations can really represent a fundamental tool, which can not only support experimental evidence, but also integrate it with a molecular-level description of the quenching processes[22,23]. In the literature, the examples of studies focusing on the possible quenching mechanisms are still limited, and almost exclusively focused on the EET mechanism[9,10,24,25]. To the very best of our knowledge, the electron-transfer mechanism has never been explored to date by atomistic simulations in real antenna complexes. In this contribution, we fill this gap, and show that electron transfer from carotenoids can be a rapid quenching pathway for excited Chl, and that it can compete with the EET to the $S_1$ state. To prove this hypothesis, we have focused on the major light-harvesting complex II (LCHII) of plants, for which previous studies have already shown the effectiveness of the EET mechanism[7,9,10,26].

LHCII has been proposed as one of the quenching sites in vivo[7,11,27]. LHCII is found in trimers, with each monomer binding 14 chlorophylls (8 Chl a, 6 Chl b) and four xanthophylls: two luteins (Lut1 and Lut2), violaxanthin (Vio), and neoxanthin (Neo)[28]. The two lutein-binding sites, L1 and L2, are related by a pseudo-twofold symmetry axis, and share a similar pigment arrangement, with one Chl (a612 in L1, a603 in L2) in close contact with the π-conjugated backbone of the Lut (Fig. 1b, c). In spite of this similarity, the two luteins show different conformations and functions[29]. The main quenching center in LHCII has been identified to be the cluster of Chls a610–a611–a612, close to the lutein L1 site[30]. In contrast, lutein in the L2 site has been associated with a light-harvesting function[31].

Starting from an all-atom molecular dynamics trajectory of trimeric LHCII in a phospholipid membrane, we have computed electron-transfer parameters for the charge separation involving lutein in sites L1 and L2, and the Chlas in close contact with them. We found that the protein environment around site L2 is not optimal for an electron transfer from the lutein to the closest chlorophyll, due to both a non-perfect arrangement of the pigments, and an electrostatically unfavorable distribution of protein residues. The situation is reversed for L1 site, for which our results suggest that the electron transfer is possible, because the CT state and the $Q_y$ state of Chl a612 are almost isoenergetic, that is, their energy difference is less than the thermal energy at room temperature. The resulting equilibrium can thus be controlled by the protein, which can tune the quenching level by changing the energy of the CT state. Finally, a kinetic model to quantify the lifetime of the complex has been built, which indicates that Lut1 can quench >90% of the Chl excitation, reducing the LHCII lifetime to <300 ps.

## Results

**Energetics and kinetics of the charge-transfer quenching.** Our analysis is based on an all-atom MD simulation of trimeric LHCII in the membrane (Fig. 1a), which allows us to sample the conformations of LHCII. In particular, we extend the MD simulation used in our previous work to 2.8 μs[24]. Our MD trajectory allows exploring the substantial flexibility of the C- and N termini, whereas the helical core domain remains rigid (Fig. 1e), in agreement with the results obtained by Liguori et al.[32] on the LHCII monomer. Overall, our MD trajectory shows a smaller flexibility than the monomer simulation of Liguori et al.[32], which is probably due to the use of trimeric LHCII instead of the LHCII monomer in this work. In particular, we found that in trimeric LHCII, the N termini are much more stable than in the monomeric simulation of Liguori et al.[32], whereas the C termini and helix D are more flexible (see SI section S2). We also note that the force field (all-atom AMBER vs. united-atom GROMOS) may play a role in the flexibility of the complex. It is in fact known that these force fields predict fairly different structural and dynamical properties for proteins[33,34].

Regarding the putative quenching sites, dimers Lut1/a612 and Lut2/a603 show low RMSD values with respect to the crystal (Fig. 1d). For these Lut/Chl dimers, our MD simulation therefore samples the crystal conformation, and does not explore other minima. This can be easily explained by recalling that, even if single-molecule experiments have shown that LHCII can switch between different conformations, these changes happen in the millisecond timescale[17,20], i.e., largely beyond what an even long conventional MD can see.

In order to test our hypothesis, we compute energies and couplings for the locally excited (LE) and CT states of the Lut–Chl pairs in L1 and L2 on 240 structures extracted from the three LHCII monomers along the MD. From these data, we estimate the driving force for charge separation in each pigment pair, and the charge-separation rate, by using Marcus theory. As shown in Table 1, the lowest CT state, corresponding to an electron moving from Lut to Chla, lies ~5000 cm⁻¹ above the $Q_y$ state of Chla. The lutein sites L1 and L2 differ significantly in the energy of the Lut⁺Chl⁻ CT state, with the CT state of the L1 dimer being some 800 cm⁻¹ (0.1 eV) lower in energy than that in the L2 dimer. Notably, this difference is mirrored in the opposite CT state (Lut⁻Chl⁺), which is much higher in energy, but displays the same difference between L1 and L2 in the reverse direction. These results suggest that the difference between the L1 and L2 sites originates in the electric field generated by the protein, which stabilizes charge separation in opposite directions

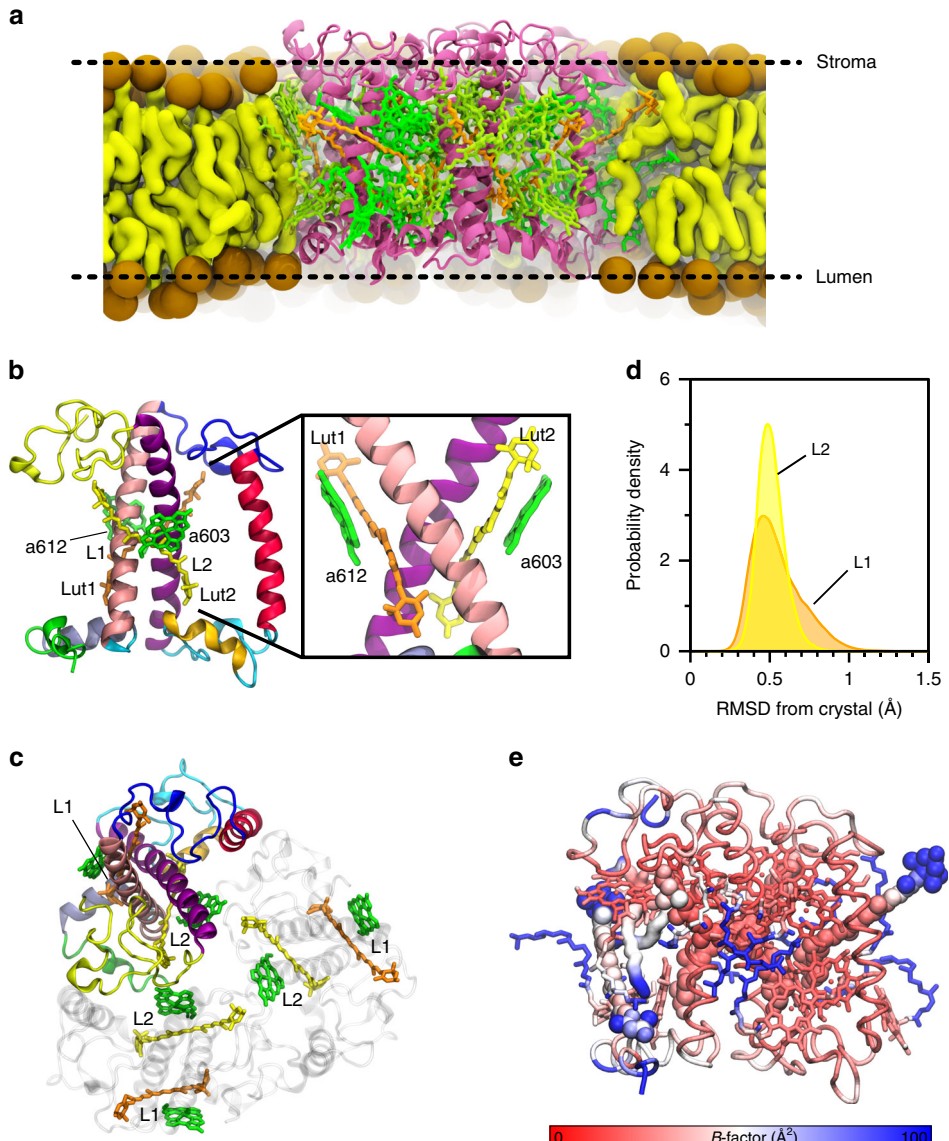

**Fig. 1 Structure and dynamics of LHCII. a** Positioning of trimeric LHCII in the phospholipid membrane. **b** View of the LHCII monomer from the membrane side, highlighting pigments of the L1 and L2 sites. The upper part corresponds to the stromal side. The inset shows a side view of sites L1 and L2. **c** View of the LHCII trimer from the stromal side, showing the pseudo-symmetrical arrangement of L1 and L2 sites. **d** Distribution of RMSD values from the crystal structure for the Lut–Chl pairs Lut1/*a612* in site L1 and Lut2/*a603* in site L2. **e** Monomer of LHCII colored by B factor (from red to blue, 0–100 Å$^2$). The protein backbone is shown as tubes, chlorophylls as sticks, and xanthophylls are shown as Van der Waals spheres, while DPPG is represented as the surface.

in sites L1 and L2. As a control, we computed the CT energies in the three monomers of the crystal structure (chains C, H, and E of the PDB). These results (Supplementary Table 3) show the same trend between L1 and L2 as computed along the MD, but with a larger difference. The reason is the lack of dielectric screening by water and the membrane, which are not included in the crystal structure. Nonetheless, the calculations on the crystal confirm that the trend we calculate is robust.

The couplings for charge separation estimated along the MD for sites L1 and L2 are ca. 250 cm$^{-1}$, with little difference between the two sites. These couplings are sufficiently large to allow for charge separation, but small compared with the reorganization energies, allowing the use of Marcus theory for estimating the charge-separation rates. While the LE–CT coupling is very similar for sites L1 and L2, the EET coupling between the bright Q$_y$ state of the Chl*a* and the S$_2$ state of lutein is significantly larger for the *a603*/Lut2 dimer of site L2, reaching almost twice its L1

counterpart. Notably, the same difference is not observed for the EET coupling of the Chl*a*'s Q$_y$ state with the S$_1$ state of lutein[24]. Nonetheless, the difference between the two sites can be traced back to the slightly different orientations of the Chl*a*/Lut dimers in sites L1 and L2[24].

The CT state has a large reorganization energy, which brings the adiabatic CT energy close to the adiabatic LE energy (see Fig. 2a). Taking the reorganization energies into account, we estimate that the CT energy minimum in site L1 is ~80 cm$^{-1}$ below the Q$_y$ state of Chl*a*, which opens a pathway for excitation quenching through charge separation. By contrast, in site L2, the CT minimum is >900 cm$^{-1}$ higher in energy than the Q$_y$ state, making the CT state thermodynamically unreachable in that site. The charge-separation rates computed with Marcus theory indicate that the CT state in the *a612*/Lut1 dimer can be populated in tens of picoseconds from the Q$_y$ state of Chl *a612*. This rate is one order of magnitude larger than the Q$_y$ → S$_1$ EET

rates determined on the same MD trajectory[24], and confirms that the conformation sampled by the present MD trajectory corresponds to a strongly quenched LHCII. We underline, however, that the exact rates of the EET mechanism strongly depend on the level of theory used to describe the Lut[35], and therefore accurate estimates for the EET mechanism are difficult to obtain. Nevertheless, a more recent study[36], based on the RASSCF transition charges from ref. [35], gives an estimate of the EET rates similar to ref. [24].

To estimate excitation quenching in our LHCII model, the mean excitation lifetime of the complex is determined by using the same coarse-grained kinetic model used in ref. [24]. In this model (Fig. 2b), Chls $a612$ and $a603$ are assumed to be in fast equilibrium with the pool of the other six Chls, and they can transfer their population to the charge-separated state involving the corresponding Lut. The charge-separated state is then assumed to recombine quickly ($\tau = 10$ ps) to the ground state or to another dark state of the Lut (the dependence of the results on this parameter is analyzed in Supplementary Fig. 6). The initial excitation is assumed to be equally partitioned between the Chls. The lifetime of the entire complex, $\tau_{complex}$, is estimated to be 277

ps, which favorably compares with the lower end of the excitation lifetime range in LHCII crystals (0.3–0.8 ns)[37]. Our kinetic model predicts that ~90% of the excitation is quenched by the Lut1$^+$ $a612^-$ CT state. In contrast, less than 3% is quenched by the CT state involving Lut2. In other words, the effect of Lut2 is trifling.

**Origin of the differences between L1 and L2 sites.** One important feature of the charge-transfer model is its ability to clearly distinguish the two lutein sites, L1 and L2, in their quenching capability. Notably, whereas both the electronic coupling between $Q_y$ and CT state and the energy of the $Q_y$ state are similar for the two sites, a marked difference in the energy of the CT state can be noticed (Table 1). The factors influencing the CT energy can be divided into (i) intramolecular coordinates of Chl$a$ and Lut, such as bond lengths and angles, (ii) intermolecular coordinates of the dimer, i.e., distance and mutual orientation, and (iii) environment. In order to disentangle these three factors, we use a multivariate linear regression on selected intramolecular and intermolecular coordinates of the Lut/Chl dimers. As intramolecular coordinates, we choose the bond-length alternation (BLA) of Chl$a$ and Lut along two different paths (Fig. 3a). The BLA is defined as the average difference between single- and double-bond lengths, and is a well-known metric of the π-conjugation. Among the possible intermolecular coordinates, we select a simplified overlap of spherical densities around the atoms Lut and Chl$a$ highlighted in Fig. 3b.

A preliminary analysis on a subset of structures in vacuo shows that more than 50% of the variability of the CT energy can be explained by a combination of the four BLAs defined in Fig. 3a (Supplementary Fig. 4). These coordinates fluctuate at the frequencies of the C=C and C–C bond stretching modes, and show no difference between the two sites. In contrast, the intermolecular coordinates are different in the two sites (Fig. 3), and they explain an additional 20% of the variability of the CT energy. The difference between L1 and L2 sites in vacuo (580 cm$^{-1}$) is fully explained by these coordinates. In site L1, lutein takes different positions with respect to the chlorine ring of the Chl$a$, and has consequently a larger overlap with the Chl$a$ electron density.

The protein environment stabilizes the CT state in both sites L1 and L2 (Supplementary Fig. 5). However, this stabilization is ~300 cm$^{-1}$ larger for L1 than for L2. This differential

---

**Table 1 Energies and couplings in the Lut–Chl dimers.**

| Parameter | $a612$/Lut1 (L1) | $a603$/Lut2 (L2) |
|---|---|---|
| E(Chl*) $Q_y$ | 15,587 ± 72 | 15,548 ± 74 |
| E(Lut*) $S_2$ | 21,114 ± 137 | 20,700 ± 135 |
| E(Lut$^+$Chl$^-$) | 20,255 ± 193 | 21,067 ± 191 |
| E(Lut$^-$Chl$^+$) | 26,027 ± 185 | 25,214 ± 173 |
| V(Chl*, Lut*) | 88 ± 185 | 176 ± 9 |
| V(Chl*, Lut$^+$Chl$^-$) | 240 ± 25 | 279 ± 18 |
| $\lambda_{CT-LE}$[a] | 5405 | 5052 |
| $\Delta G_{CT-LE}$[a] | −82 | 951 |
| $k_{CS}$(ns$^{-1}$)[b] | 34.0 ($T = 29$ ps) | 4.9 ($T = 205$ ps) |

Average values and 95% confidence intervals of energies (E) and couplings (V), obtained from the sampling of 80 snapshots for each of the three monomers. Coupling averages are given as root mean square (RMS), instead of arithmetic mean. Reorganization energies ($\lambda$) and driving forces ($\Delta G$) are also given, as estimated from the variance of the CT and LE energies. All values are in cm$^{-1}$.
[a]These quantities refer to the lowest CT/LE states, i.e., chl* and lut$^+$chl$^-$.
[b]Estimated rate/time of charge separation.

---

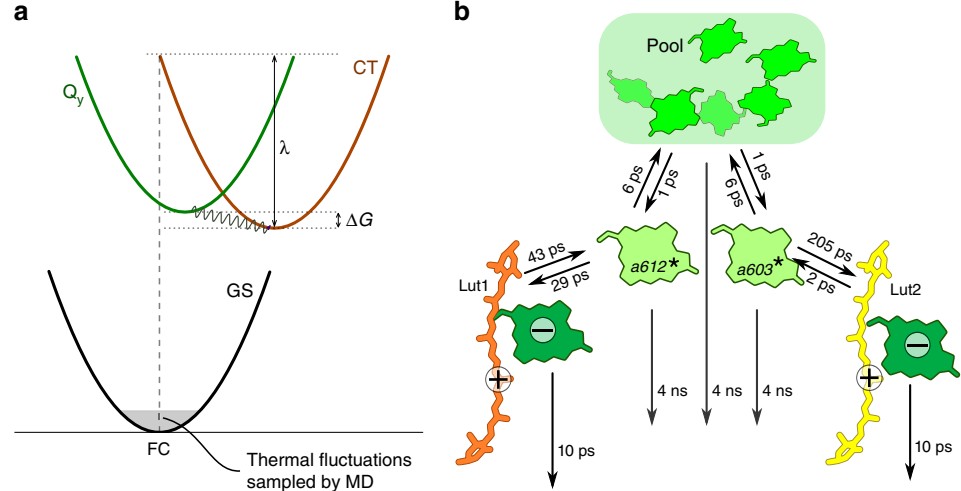

**Fig. 2 Model of charge-separation quenching in LHCII. a** Scheme of charge separation in the Lut-Chl a dimers, highlighting the relevant parameters that determine the charge-separation rate. The energy minima of $Q_y$ and CT states, and the reorganization energy $\lambda$, are estimated from the energy fluctuations along the MD. **b** Coarse-grained kinetic model of excitation quenching. Numbers beside arrows indicate the inverse rate of the elementary steps. Downward-pointing arrows indicate irreversible decay to the ground state.

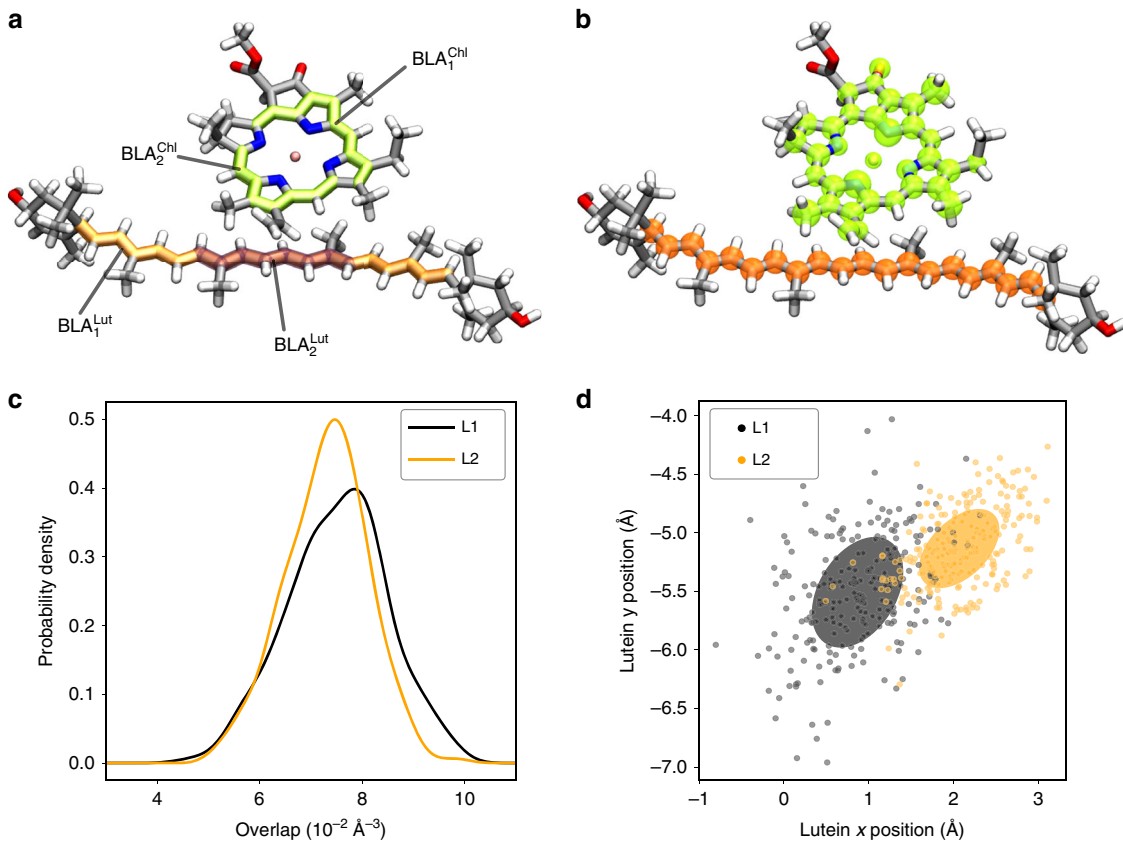

**Fig. 3 Geometrical analysis of Lut–Chl pairs.** Definitions of (**a**) BLAs and (**b**) atoms used for the density overlap. **c** Distribution of overlap values in sites L1 and L2. **d** Position of Lut1 (Lut2) in a *xy* reference frame defined by the chlorine ring of Chl *a*612(*a*603).

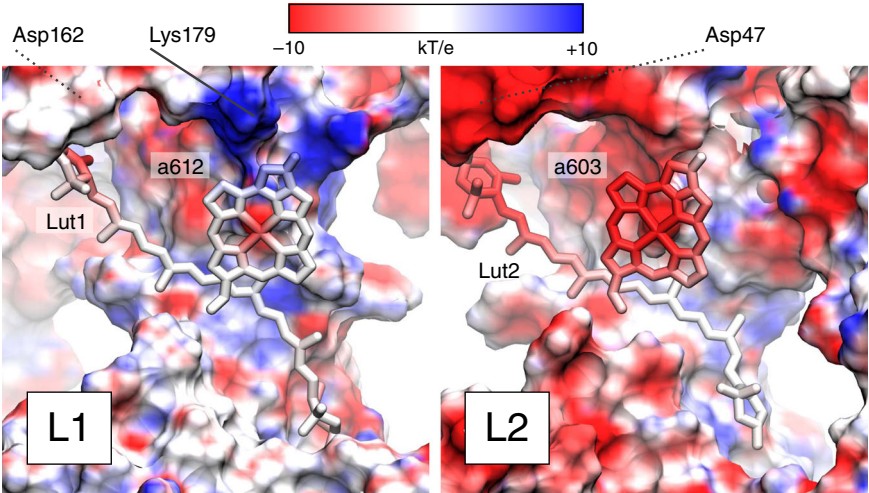

**Fig. 4 Electrostatic potential around the L1 (left) and L2 (right) sites of LHCII.** Only the protein surface is shown, together with the Chl–Lut dimers, both colored by electrostatic potential. The electrostatic potential was computed on one of the MD frames by using the APBS program[60]. The solid and dotted lines indicate the position of some charged residues close to the dimers on the stromal side.

environment effect adds to the difference originating from the relative orientation of Chl and Lut in the two sites, and can be traced back to the charge distribution of the protein residues around the Lut/Chl dimers. In fact, the electrostatic potential in the two sites is very different, as seen in Fig. 4.

Site L2 is located toward the center of the LHCII trimer, close to a negative potential region around the N terminus, to which Lut2 is bound. Chl *a603* is located in a region of negative potential (see Fig. 1c), which tends to destabilize the Lut → Chl electron transfer. In contrast, site L1 is on the outer side of the

trimer, and Lut1 is bound to the stromal loop. Chl *a612* is close to a region of positive potential, and in particular to the positively charged Lys179. This explains the stronger stabilization of the CT state in site L1.

From the above analysis, it seems clear that the protein environment around site L2 is optimized to disfavor an electron transfer from Lut2 to Chl *a603*, both in terms of a slightly different arrangement of the pigments, and of a distribution of protein residues. Recent two-dimensional spectroscopy experiments have shown that in plant LHCII, Lutein 2 is optimized for

light harvesting, rather than for excitation quenching[31]. Notably, the bright excitation of Lut2 is red-shifted in trimeric LHCII with respect to Lut1[31,38], which is reproduced by our calculations (Table 1), together with a larger Lut–Chla coupling in site L2. Taken together, these results show that our atomistic description of trimeric LHCII is capable of discriminating between the two lutein sites, only one of which is optimized for excitation quenching.

## Discussion

Although, up to now, the involvement of lutein in a CT-based quenching mechanism has been seldom considered for LHCII[14], our calculations show that a CT state between Lut1 and Chl *a612* is able to quench the excitation of the entire LHCII complex. CT-based quenching was first proposed for minor antenna complexes[39], also on the basis of the spectral features of the zeaxanthin radical cation[13]. However, later evidence showed that zeaxanthin is not directly involved in excitation quenching in LHCII, but could possibly act as an allosteric modulator, influencing the conformation of LHCII[40,41]. We underline that the role of lutein in CT quenching was proposed for the minor antenna complex CP26, where a band characteristic of the Lut radical cation was observed at 940 nm in the transient absorption spectra[42]. In contrast, the band at 980 nm was assigned to zeaxanthin[13,42]. A lutein radical cation feature was not observed in transient absorption spectra of LHCII. This can be explained by the fast recombination of the charge-separated state, which effectively quenches the excitation without allowing for a substantial population of the charge-separated state. Therefore, the absence of a radical cation band in the transient absorption spectra does not necessarily imply the absence of a charge-separation mechanism. Indeed, Stark fluorescence spectroscopy in LHCII trimers devoid of zeaxanthin showed that external electric fields decreased the fluorescence intensity by accelerating nonradiative processes in the antenna[14]. This could only be explained by a CT state being directly involved in the quenching mechanism.

Recently, Mascoli et al. have observed a spectral feature associated with a quencher state in CP29[43]. This feature resembles that of a Car triplet state, albeit with a much smaller lifetime, and was assigned to the debated $S^*$ state, implying that the Chla quenching in CP29 arises from EET to the $S^*$ state. The nature of the $S^*$ state remains to be completely understood[44]; in the same study, the authors suggested that such a state can be ascribed to a $S_1$ state having a different torsional conformation. From the data here obtained, we could alternatively suggest that the spectral signature observed by Mascoli et al. is the result of a charge recombination to the hot ground state of lutein, or even to the $S_1$ state. In fact, the $S_1$ state is close in energy to the $Q_y$ state of Chla, and therefore would be near isoenergetic also to the CT state, promoting charge recombination.

Our results also indicate that the energy difference between CT state and the $Q_y$ state of Chl *a612* is less than the thermal energy at room temperature. The resulting equilibrium can thus be controlled by the protein, which can tune the quenching level by changing the energy of the CT state. As we have shown above, this tuning can be exerted by the protein, either by modulating the electric field around the carotenoid, or by changing the position and relative orientation of Chla and Lut. In order to investigate the role of the protein in changing the Chl *a612*—Lut1 dimer arrangement, we show in Fig. 5 the two configurations of the L1 site that show, respectively, the largest and smallest overlap values among all the considered dimer structures. The overlap values of these structures differ by ~$5 \times 10^{-2}$ Å³. Based on the regression coefficients (Supplementary Table 4), we can estimate an ~2000 cm⁻¹ difference in the CT energy, solely due to the change in overlap between these extremes.

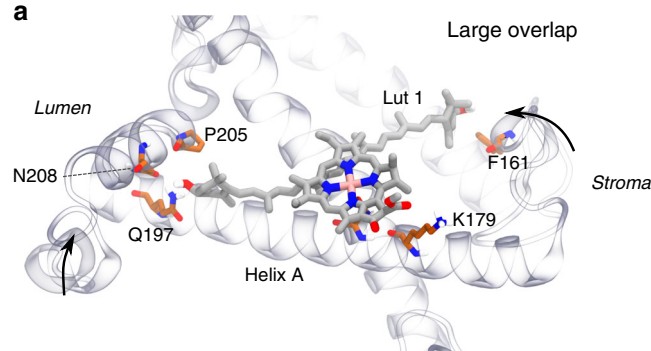

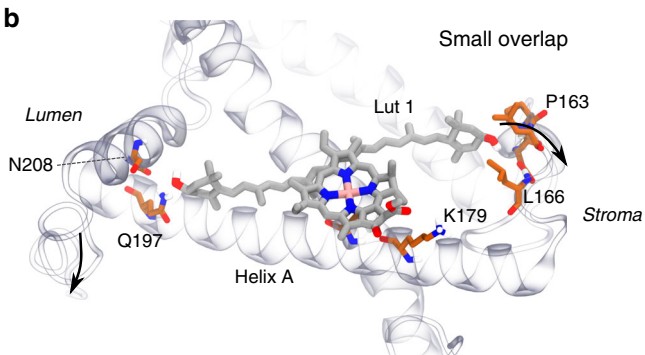

**Fig. 5 Arrangement of the L1 site in extreme overlap configurations. a** Large overlap and (**b**) small overlap. The overlap values are, respectively, $9.8 \times 10^{-2}$ and $4.7 \times 10^{-2}$ Å³. The residues close to the terminal rings of Lut are shown for each case; residue names are given in the one-letter code. The arrows show the movement of the stromal loop with respect to the L1 pocket.

Although the two structures in Fig. 5 are very similar in the helical part, they show some differences in the stromal loop and in the C terminus. In particular, in the configuration with the largest overlap, the C terminus is more compact and closer to Helix A. Also, the stromal loop is found closer to the core of the complex, and the lutein terminal ring is displaced further from Helix A. These results point to a role of the most mobile regions exposed to the solvent (Helix D and the stromal loop, see also Supplementary Fig. 2) in tuning the arrangement of lutein within the L1 site. The differences observed between Lut1 and Lut2 demonstrate that a rather limited rearrangement in the relative orientation of the Lut1-*a612* pair could raise the energy of the CT state above that of the $Q_y$ state of Chl *a612*. Raising the free energy of charge separation to ~700 cm⁻¹ is sufficient to recover the 2-ns lifetime observed in photosynthetic membranes[45] (Supplementary Fig. 6). Based on the arrangement of Lut and Chla in sites L1 and L2, we propose that a displacement of ~1 Å of the lutein could raise the energy of the CT state enough to shut down the CT quenching.

Several studies have hypothesized that chlorophyll–chlorophyll CT states are responsible for excitation quenching in LHCII, as well as other spectral signatures[46,47]. Therefore, it is worthwhile to investigate whether a Chl–Chl charge separation can also quench the Chl excitation. To check this hypothesis, the calculations performed for Lut–Chl dimers have been repeated for the strongly interacting *a612/a611* and *a603/b609* chlorophyll dimers, located respectively near the L1 and L2 sites. The results (Supplementary Table 2) show that, in both of these dimers, the CT state is more than 800 cm⁻¹ (0.1 eV) above the locally excited $Q_y$ state of Chl *a611(a603)*. As expected, the CT state is somewhat lower in energy in the *a603/b609* heterodimer with respect to the homodimer found in site L1.

While the occurrence of charge separation in these Chl–Chl dimers can be ruled out on the basis of the calculated energies, we need to consider the possibility that these CT states mix with the exciton states of LHCII, affecting their energy. In particular, mixing of high-lying CT states with exciton states redshifts the exciton energy[48,49]. Therefore, Chl–Chl CT states have been identified as responsible for red-shifted bands in the fluorescence spectra of LHCII[50,51] and other plant antenna complexes[52,53]. These "red states" are likely not directly related to excitation quenching[54], but they can be seen as spectral signatures of quenched conformations[55]. The involvement of CT mixing in these states is substantiated by the large exciton–phonon coupling measured in fluorescence[47]. Since these states cannot participate in excitation quenching, they might have a different role in regulating the energy flow through LHCII, acting as excitation energy traps.

We have demonstrated that a charge-separation mechanism for excitation quenching involving lutein is possible in LHCII. Ours is the first study of this type, integrating, in a single computational strategy, fully atomistic molecular dynamics and multiscale quantum chemical descriptions, which include electrostatic and polarization effects of the protein and the composite environment. Precisely, thanks to this detailed description, we have also unveiled the structural features that make the two putative quenching sites different, and we have shown that very small changes in the structural constraints and the electric field induced by the protein can indeed switch on and off the quenching mechanism. Let us highlight that simulations of the EET mechanism predict that both luteins participate in quenching[10,24]. On the contrary, the charge-separation mechanism presents an exquisite sensitivity to the arrangement of lutein and chlorophyll, and therefore predicts L1 as the primary quenching site.

We have, however, to observe that our results are based on structures obtained by sampling the region of space close to the crystal structure[56]. It is generally accepted that the crystal structures of LHCII[56,57] are representative of a quenched conformation[58], with reduced fluorescence timescales (0.3–0.8 ns) compared with in vivo membranes[37]. The conformation of LHCII in the crystal is thought to be similar to that obtained upon aggregation of LHCII trimer[11], whereas the in vivo conformation remains unknown. Indeed, experimental and computational results suggest that LHCII explores several conformations in solution[17,32]. However, little is known about the behavior of the antenna in the photosynthetic membrane, which has been compared with the detergent environment only in recent experiments[26]. In order to understand the in vivo quenching mechanism of LHCII, the unquenched conformations of LHCII in the membrane need to be thoroughly characterized by a joint computational and experimental effort. Further work in this direction is being carried out in our group.

## Methods

**Quantum chemical calculations**. We extracted 80 snapshots every 10 ns along the MD trajectory. From these snapshots, we prepared QM calculations for each of the three Lut/Chl dimers equivalent by symmetry, for a total of 240 calculations for each considered dimer. Excited states were computed at the TDA/ωB97X-D/6-31+G(d) level of theory, including the effect of the polarizable environment through MMPol. The CT energies and couplings were computed through our multi-FED–FCD diabatization scheme[49], whereas the energy of the Chl state was recomputed at the full TD-DFT/ωB97X-D/6-31+G(d) level (i.e., without the Tamm–Dancoff approximation). All excited-state calculations were performed with a locally modified version of the Gaussian package[59]. In all QM calculations, the phytyl tail of the chlorophyll was cut after the first aliphatic carbon and kept in the MM region. While TD-DFT performs very well for the LE states of chlorophylls, whose excitation energy can be compared with experiments, it has difficulties in dealing with CT states, for which experimental data are missing. Therefore, the quality of the TD-DFT description of CT states was assessed through a benchmark at the CC2 and ADC(2) levels of theory, which is detailed in the Supplementary Methods.

**Marcus parameters and rates**. The remaining parameters used in the Marcus rate equations, driving force and reorganization energy, were estimated based on the energy fluctuations of LE and CT states. The adiabatic energy of each state was computed as $\Delta G_X = \langle E_X \rangle - \frac{\sigma_X^2}{2k_B T}$ ($X = $ CT/LE), where $k_B T$ is the thermal energy, $\langle E_X \rangle$ is the average, and $\sigma_X^2$ is the variance of the energy of state $X$. The driving force is then obtained as the difference between the two adiabatic energies. The reorganization energy $\lambda$ for the LE/CT radiationless transition was similarly determined as $\frac{\sigma_{\Delta E}^2}{2k_B T}$, where $\Delta E$ is the energy difference between the two states.

## Data availability

The data that support the findings of this study are available from the corresponding authors upon request. The source data underlying Figs. 1d, 3c–d, Table 1, Supplementary Figs. 1–7, and Supplementary Tables 1–4 are provided as a Source Data file.

## Code availability

The custom code used for this study is available from the corresponding authors upon request.

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

## Acknowledgements

B.M. and L.C. acknowledge funding by the European Research Council, under the grant ERC-AdG-786714 (LIFETimeS). D.J. thanks the CCIPL computing center installed in Nantes for generous time allocation.

## Author contributions

L.C. and D.C. performed TD-DFT calculations; D.J. performed benchmark quantum chemical calculations; L.C. performed data analysis; L.C. and B.M. designed the research; L.C., B.M. and D.J. wrote and edited the paper. All authors approved the final version of the paper.

## Competing interests

The authors declare no competing interests.
