## [Peer Review File · Nature Communications]

Reviewers' comments:

Reviewer #1 (Remarks to the Author):

This manuscript aims to discover the process of nonphotochemical quenching by lutein in the lutein/chlorophyll pairs in LHCII. The authors proposed and justify that quenching of the Chl-a excited state by Lutein occurs via electron transfer from Lutein, rather than the more commonly accepted mechanism of EET from Chl-a to Lutein. This is a very interesting and powerful inquiry.

From a methodological point of view, this work is at the forefront of multiscale QM/MM simulations. Before further consideration, I would like the authors to address some questions.

1) Doing QM (or QM/MM) calculations on snapshots taken from an MD simulation at the purely MM level is quite impressive. If I may be honest, it rarely works. The force field energy surface is usually so uncorrelated with the QM/MM energy surface that it is almost impossible to get good statistics. On top of that, the MM-MD does not necessarily sample the QM/MM basin around the QM/MM minimum. The authors do in fact assume this to calculate the average excitation energy and the Marcus theory parameters out of the MD energy fluctuations.

Reading carefully the SI, I do see that authors use a customized force field for Lutein, so it is possible that their force field is extraordinarily good. I would be convinced of this if authors could show a histogram of the fluctuations of each of the 4 excitation energies they show in Table 2. Those confidence intervals are amazingly small. I would like to see how those intervals compare with the standard deviation of the excitation energy fluctuations.

I am asking all this because the entire justification of their hypothesis is based on the exact order of the excitations.

2) I did a quick TDA calculation of Lutein in Gaussian at the wB97XD/6-31+g* with scrf=(pcm, benzene) and got the S1 at 512 nm ($f=0.245$) and S2 at 493 nm ($f=4.3$) (similar results with ethanol as a solvent). That is good because it means that the theory gives the correct S1/S2 order. So, I am probably missing something here but why the "dark" S1 state is not being considered? Could S1 be also a sync for the Qy excitation? Wouldn't S1 lie in between Qy and S2, possibly competing with the E(Lut+Chl-) CT excitation to quench Qy?

This paper by Polivka <https://doi.org/10.1016/j.bbabi.2016.09.001> shows that S1 can also be a channel of quenching.

Have the authors considered calculating the average S1 excitation to see if it has oscillatory strength at finite temperature?

3) How would the excitations change at zero temperature (i.e. QM/MM calculations directly from the X-ray structure as is, or partially minimized)? I mean, is MD fundamentally necessary to discern the quenching process?

4) Provided we accept table 2, is it correct to assume that quenching of a603 is via EET to Lut2? That is, there would be two mechanisms, one via CT and the other via EET?

Reviewer #2 (Remarks to the Author):

The manuscript by Mennucci's group reports on a combination of molecular dynamics simulations, quantum chemical calculations, and kinetic modeling for a major antenna LHCII trimer model embedded within a thylakoid membrane mimetic (DOPC lipid patch). The main conclusion of the study is novel and suggests that the Non-Photochemical Quenching (NPQ) of chlorophyll fluorescence in LHCII under photoprotection, relates to an electron transfer from Lutein-1 (Lut1) to

Chl-a 612. This leads to the formation of a short-lived charge transfer (CT) state, instead of an exciton energy transfer (EET) from Chl-a 612 to the Lut1 short-lived dark states for the quenching. In case this is indeed the case, the study is extremely important to scientists working in Photosynthesis and especially the NPQ field. The authors also report on the possible protein scaffold dynamics that can tune the CT state energetics. Thus, given the structural homology between LHCII and other LHCS from various species, this study can be of interest to other related disciplines within the light harvesting field. The conclusions certainly advance our understanding and can influence thinking in the field. However, in my opinion, there are several (and major) issues that need to be addressed prior to the potential publication in Nat Comm.

Major issue:

(1) The authors indicate that the EET rates between Chl-a 612 Q_y / Lut1 S1 dark state reported in ref 24 are considerably smaller compared to the rates calculated for the Chl-a/ Lut1 CT state in the current study. However, in my opinion the comparison could be problematic, and thus the conclusion that the CT state is more favorable, than the EET mechanism under NPQ, could be severely weakened. The trajectory in ref 24 is considerably shorter than the one used in the current study. More importantly, the CT energies and couplings are computed through a multi-FED-FCD diabaticization scheme (ref 46) in the current study based on DFT. The description of couplings in CT states involving carotenoids by DFT could be already problematic. The associated transition density produced might not be realistic, as it comes from orbitals that are physically meaningless basis functions (Biophys. Chem. 2019, 246: 16-24). On the other hand, the absolute couplings-rates reported for EET (Q_y to S1) in ref 24 are based on coulombic coupling from transition densities calculated at the AM1 level. For the latter calculations there is a scaling factor involved in order to 'reproduce' a small transition dipole moment for the Lut S1. In addition the couplings for CT/ EET could be increased (or decreased) if a different level of theory was to be employed. In fact, employing the RASSCF level of theory for the transition densities, the excitonic couplings between Chl-a/ Lut1 are increased, compared to the AM1 level (Biophys. Chem. 2019, 246: 16-24). In my opinion, the couplings/ rates are only meaningful as shifts even within the same calculation scheme, and not as absolute numbers. Thus, comparing these numbers based on completely different approaches (i.e. level of theory) is questionable for the physical mechanism probed.

Other issues:

(2) The MD simulations of ref (24) have been extended to approximately 3.0us in the current study. I assume that the standard protonation states for the LHCII residues reported in ref (24), are employed also for this study, and both were defined based on pH neutrality at the lumenal-stromal sides of LHCII. The protonation state of residues however, might have some consequences on the environment of mainly Lut1, and especially the electrostatic potential as reported in Figure 4. This has to be clarified, especially in the case where NPQ is assumed to be triggered by a transthylakoid membrane Δ pH, or by the interaction with PsbS, that could potentially also affect residue protonations in the vicinity of Lut1.

(3) Liguori et al. (ref 31) reports that there are large variations at the Chla611-Chla612-Lut1 cluster of LHCII, even at shorter time scales, compared to the crystal structure. Is this compatible with the small RMSD values for the Chl-a 612/ Lut1 pair, with respect to the crystal structure conformation, reported in the current study? The Chl-a 612/ Lut1 and Chl-a 603/ Lut2 sites are located within totally different domains of the LHCII protein scaffold. Chl-a 603/ Lut2 is located close to the monomer-monomer intersections where fluctuations are considerably limited, whereas the Chl-a 612/ Lut1 is more or less (partly) exposed to the periphery of the LHCII trimer. Thus, the latter could exert more fluctuations. Moreover, a recent study on the LHCII trimer (J. Phys. Chem. B 2019, 123, 45, 9609-9615) reports also fluctuations on the relative helix-A/D orientations in the transition from the light harvesting to the quenched state. Are these fluctuations consistent with what it is observed in the current study?

(4) Are all the LHCII monomers in the model sampling the same conformations at the same time?

Or is there a heterogeneity within the trimer?

(5) How far apart are the frames out of the MD trajectory, used for the QM/MM calculations and the subsequent calculation of the mean excited life times? Can we talk about actual lifetimes of the excited state of the LHCII complex, if the input data are separated considerably in time?

(6) The authors claim that the CT state they have characterized within LHCII is able to reduce the LHCII lifetime to less than 300 ps. Can't this be achieved by the EET scheme? The crystal structure of LHCII is highly quenched, which gives high coupling values also between Chl-a 612 Qy/ Lut1 S1 states (Biophys. Chem. 2019, 246: 16-24; J. Phys. Chem. B 2019, 123, 45, 9609-9615).

Minor issues:

(7) I am not sure I understand the comment on page 3 "To the very best of our knowledge, the electron-transfer mechanism has never been explored to date for real antenna complexes". Do the authors mean the electron transfer between Chl-a/ Lutein within an antenna complex? As such mechanism has been proposed between Chl/ Zeaxanthin in antenna complexes (Science. 320 (2008) 794–797; Science. 307 (2005) 433–436).

(8) Concerning the electrostatic potentials in Fig. 4, how are these calculated? Are these based on one frame (maybe the crystal), or multiple frames? As mentioned also in comment #2 the Chl-a 612/ Lut1 environment could be highly vulnerable to also protein-protein interactions (LHCII aggregation, or LHCII-PsbS).

(9) What are the designated residues at the lumen-stroma sides of LHCII in Figure 5? The authors have indicated some residues by solid cylinders, but no reference of their identity is given.

(10) The study by Mennucci's group reports fluctuations of the same state of the LHCII (quenched) over the MD trajectory. In Figure S5 the lifetimes reported cannot be mapped on actual sampled structures of LHCII along the MD trajectory, but these are produced by arbitrarily changing the charge-recombination rate to the ground state. However, the structures in Figure 5 depict L1 site in large and small overlap configurations. What would be the change in the couplings, charge-recombination rates & ultimately lifetimes of the LHCII complex for these structures in Figure 5?

Reviewer #3 (Remarks to the Author):

The manuscript by Cupellini et al. represents the results of a very thorough theoretical investigation of the mechanism of chlorophyll excited state dissipation via carotenoids of LHCII complex. The authors derived theoretically for the first time that the charge transfer mechanism is likely to be involved in this process. All previous theoretical studies and some experimental ones were focused upon EET mechanism. Hence this work is of general interest and fundamentally novel contribution to the field of the regulation of light harvesting in the photosynthetic antenna - NPQ. Indeed, only zeaxanthin in the minor LHCII antenna complexes has been proposed as a pigment that could be involved in NPQ. However, the recent research has clearly demonstrated that the quenching in LHCII complex *in vivo* does not involve zeaxanthin. Moreover, *in vitro* LHCII devoid of zeaxanthin could undergo large reversible quenching. Therefore, the understanding of the physics and identity of the quencher in LHCII is of great significance. Cupellini et al., not only identifies the quencher and the mechanism of quenching they also explained the reasons behind lutein 1 not lutein 2 being a quencher. This is a very significant insight into the atomic environment of lutein 1 that makes it possible the energy quenching via charge transfer mechanism. Interestingly the transfer times to this carotenoid are consistent with the previously expressed notion that the NPQ quencher is rather slow, hence the economic nature of the regulation of light harvesting in plants, i.e., when the reaction center is actually closed, then NPQ starts to work, but when it is open, it gains priority for the captured energy.

Reviewer #1:

This manuscript aims to discover the process of nonphotochemical quenching by lutein in the lutein/chlorophyll pairs in LCHII. The authors proposed and justify that quenching of the Chl-a excited state by Lutein occurs via electron transfer from Lutein, rather than the more commonly accepted mechanism of EET from Chl-a to Lutein. This is a very interesting a powerful inquire.

Authors' Reply: We sincerely thank the Reviewer for the positive comments.

From a methodological point of view, this works is at the forefront of multiscale QM/MM simulations. Before further consideration, I would like the authors to address some questions.

1) Doing QM (or QM/MM) calculations on snapshots taken from an MD simulation at the purely MM level is quite impressive. If I may be honest, it rarely works. The force field energy surface is usually so uncorrelated with the QM/MM energy surface that it is almost impossible to get good statistics. On top of that, the MM-MD does not necessarily sample the QM/MM basin around the QM/MM minimum. The authors do in fact assume this to calculate the average excitation energy and the Marcus theory parameters out of the MD energy fluctuations. Reading carefully the SI, I do see that authors use a customized force field for Lutein, so it is possible that their force field is extraordinarily good. I would be convinced of this if authors could show a histogram of the fluctuations of each of the 4 excitation energies they show in Table 2. Those confidence intervals are amazingly small. I would like to see how those intervals compare with the standard deviation of the excitation energy fluctuations. I am asking all this because the entire justification of their hypothesis is based on the exact order of the excitations.

Authors' Reply: We agree with the Reviewer that, in principle, QM/MM calculations on MD structures should be performed with care as such a protocol can lead to artefacts. This is especially true for carotenoids, whose excitations have large vibronic coupling and thus are more sensitive to the internal geometry of the pigment. Indeed, the force field we use here for carotenoids (Prandi et al., *J Comput Chem* (2016), **37**, 981) was designed and parameterized specifically to deal with this problem. Our Lutein force field is designed to reproduce not only QM bond lengths, but also the normal modes of the isolated pigment. In addition, the excitations computed on top of the force-field geometries were in agreement with those computed on the QM optimized structures, both in terms of energy ordering, which is crucial as pointed out by the Reviewer, and also quantitatively.

The 95% confidence intervals for the mean were computed from the critical values of the Student's t distribution and from the standard error of the mean. Given that we are using 240 different calculations (for each site L1 or L2), these confidence intervals roughly correspond to $\sim 1/8$ of the sample standard deviation. The largest standard deviation of the excitation energy is found for the CT state ($\sim 1500 \text{ cm}^{-1}$), which then gives a value of $\sim 200 \text{ cm}^{-1}$ for the confidence interval as reported in Table 1.

2) I did a quick TDA calculation of Lutein in Gaussian at the wB97XD/6-31+g* with scrf=(pcm, benzene) and got the S1 at 512 nm (f=0.245) and S2 at 493 nm (f=4.3) (similar results with ethanol as a solvent). That is good because it means that the theory gives the correct S1/S2 order. So, I am probably missing something here but why the “dark” S1 state is not being considered?

Could S1 be also a sync for the Qy excitation? Wouldn't S1 lie in between Qy and S2, possibly competing with the E(Lut+Chl-) CT excitation to quench Qy? This paper by Polivka <https://doi.org/10.1016/j.bbabi.2016.09.001> shows that S1 can also be a channel of quenching. Have the authors considered calculating the average S1 excitation to see if it has oscillatory strength at finite temperature?

Authors' Reply: It is true that TDA-DFT gives a “dark” state below the bright state with some functional/basis set combinations. However, we wish to point out that in previous studies (Spezia, et al. *Phys Chem Chem Phys* **19**, 17156–17166 (2017); Andreussi, et al. *J. Chem. Theory Comput.* **11**, 655 (2015)) we have used DFT/MRCI to show that the energy position of this low-energy state is due to cancellation of the effects that the neglect of the multideterminant character has on the ground and the excited states (this also explains why TDA gives a better picture than TDDFT). This finding tells us that the TDA description cannot be used as a robust method for describing the S1 dark states of carotenoids, as this fortuitous cancellation of effects strongly depend on the structure and the environment. In particular, within the highly structured and anisotropic environment of LHCII, our TDA wB97X-D level of theory yields a bright state as the first lutein state and this is true for all the different configurations we have obtained from MD.

We are aware that the leading interpretation of the quenching mechanism in LHCs is that the Chl transfers energy to the S1 state of Lut. This is why we compare our charge separation rates with the EET rates computed in Ref. 24 on the same molecular dynamics we are using here. Clearly, we cannot exclude that the EET mechanism competes with the charge separation. However, we pointed out in the Discussion that all models of the EET mechanism predict that both luteins quench the excitation of the chlorophylls, which is in contradiction with experimental evidence.

3) How would the excitations change at zero temperature (i.e QM/MM calculations directly from the Xray structure as is, or partially minimized)? I mean, is MD fundamentally necessary to discern the quenching process?

Authors' Reply: we thank the Reviewer for this question. One of the main reasons that compelled us to use structures from a MD trajectory is that the charge-transfer couplings (and energies) are strongly dependent on the distances between molecules (this was demonstrated for bacteriochlorophyll dimers in ref. 45). In addition, averaging over a MD trajectory allows us to take into account the heterogeneous environment (water and membrane) of LHCII in a rigorous way.

Following the comment of the Reviewer, we have now computed energies and couplings on the three monomers of the crystal structure and added the results in the Supporting Information (Table S3). The results show a similar trend between CT energies as seen along the MD trajectory, albeit with larger differences between L1 and L2. This effect can be ascribed to the lack of solvent and membrane, which partially screen the protein electric field. Notably, the CT couplings are very different from one another, and different from the average of the MD trajectory. This is not surprising considering the sensitivity of the CT couplings to the distances and relative orientations between pigments. We added the following sentences to the main text:

“As a control, we computed the CT energies in the three monomers of the crystal structure (Chains C, H, E of the PDB). These results (Table S3) show the same trend between L1 and L2 as computed along the MD, but with a larger difference. The reason is the lack of dielectric screening by water and membrane, which are not included in the crystal structure. Nonetheless, the calculations on the crystal confirm that the trend we calculate is robust.”

4) Provided we accept table 2, is it correct to assume that quenching of a603 is via EET to Lut2? That is, there would be two mechanisms, one via CT and the other via EET?

Authors' Reply: As stated above, we cannot exclude the EET mechanism within either the L1 or the L2 site. In fact, the values reported in Table 1 for the Chl*/Lut* exciton coupling refer to the S2 state of Lutein and are not necessarily related to the EET coupling involving the S1 state. Given the experimental evidence of Lut1 being associated with quenching, the EET mechanism should involve mainly Chl a612 and Lut1. However, most of the theoretical determinations of the EET quenching give essentially the same result for sites L1 and L2, which hints that EET is, at best, not the sole mechanism.

Reviewer #2:

The manuscript by Mennucci's group reports on a combination of molecular dynamics simulations, quantum chemical calculations, and kinetic modeling for a major antenna LHCII trimer model embedded within a thylakoid membrane mimetic (DOPC lipid patch). The main conclusion of the study is novel and suggests that the Non-Photochemical Quenching (NPQ) of chlorophyll fluorescence in LHCII under photoprotection, relates to an electron transfer from Lutein-I (LutI) to Chl-a 612. This leads to the formation of a short-lived charge transfer (CT) state, instead of an exciton energy transfer (EET) from Chl-a 612 to the LutI short-lived dark states for the quenching. In case this is indeed the case, the study is extremely important to scientists working in Photosynthesis and especially the NPQ field. The authors also report on the possible protein scaffold dynamics that can tune the CT state energetics. Thus, given the structural homology between LHCII and other LHCS from various species, this study can be of interest to other related disciplines within the light harvesting field. The conclusions certainly advance our understanding and can influence thinking in the field. However, in my opinion, there are several (and major) issues that need to be addressed prior to the potential publication in Nat Comm.

Authors' Reply: We sincerely thank the Reviewer for the positive comments.

Major issue:

(1) The authors indicate that the EET rates between Chl-a 612 Q_y / LutI S₁ dark state reported in ref 24 are considerably smaller compared to the rates calculated for the Chl-a/ LutI CT state in the current study. However, in my opinion the comparison could be problematic, and thus the conclusion that the CT state is more favorable, than the EET mechanism under NPQ, could be severely weakened. The trajectory in ref 24 is considerably shorter than the one used in the current study. More importantly, the CT energies and couplings are computed through a multi-FED-FCD diabaticization scheme (ref 46) in the current study based on DFT. The description of couplings in CT states involving carotenoids by DFT could be already problematic. The associated transition density produced might not be realistic, as it comes from orbitals that are physically meaningless basis functions (Biophys. Chem. 2019, 246: 16-24). On the other hand, the absolute couplings-rates reported for EET (Q_y to S₁) in ref 24 are based on coulombic coupling from transition densities calculated at the AM1 level. For the latter calculations there is a scaling factor involved in order to 'reproduce' a small transition dipole moment for the Lut S₁. In addition, the couplings for CT/ EET could be increased (or decreased) if a different level of theory was to be employed. In fact, employing the RASSCF level of theory for the transition densities, the excitonic couplings between Chl-a/ LutI are increased, compared to the AM1 level (Biophys. Chem. 2019, 246: 16-24). In my opinion, the couplings/ rates are only meaningful as shifts even within the same calculation scheme, and not as absolute numbers. Thus, comparing these numbers based on completely different approaches (i.e. level of theory) is questionable for the physical mechanism probed.

Authors' Reply: Our choice of level of theory was motivated by the best possible compromise between computational cost and accuracy. We would like to stress that our DFT calculations are based on a long-range corrected DFT functional with correct asymptotic behavior, which is able to describe the CT excitations irrespective of the separation between the donor and acceptor moieties. Moreover, our level of theory is benchmarked against more accurate (and self-interaction free) levels of theory (CC2 and ADC(2)). As we have already commented in the answer to question 2) of Reviewer 1, the requirements for correctly describing the EET to the SI state of Lutein are different, and TD-DFT is clearly not suited for this task. Nonetheless, we have chosen to compare our charge separation rates with the EET rates of ref. 24 especially because they are computed on the same MD trajectory.

We do understand that there is a significant uncertainty associated with the determination of Qy/SI EET couplings. However, in our opinion, it is important to point out this difference between EET and CT rates, because the EET rates in ref. 24 are based on the same MD as the present study. Moreover, the data reported in the Supporting Information of *J. Phys. Chem. B* 2019, 123, 45, 9609-9615, which are based on the RASSCF TrEsp charges, show inverse EET rates in the order of 200 ps, i.e., similar to Ref. 24. For the sake of clarity, we added the following sentence:

“We underline, however, that the exact rates of the EET mechanism strongly depend on the level of theory used to describe the Lut,³⁵ and therefore accurate estimates for the EET mechanism are difficult to obtain. Nevertheless, a more recent study,³⁶ based on the RASSCF transition charges from Ref. 35, gives an estimate of the EET rates similar to Ref. 24.”

Other issues:

(2) The MD simulations of ref (24) have been extended to approximately 3.0us in the current study. I assume that the standard protonation states for the LHCII residues reported in ref (24), are employed also for this study, and both were defined based on pH neutrality at the luminal-stromal sides of LHCII. The protonation state of residues however, might have some consequences on the environment of mainly LutI, and especially the electrostatic potential as reported in Figure 4. This has to be clarified, especially in the case where NPQ is assumed to be triggered by a transthylakoid membrane Δ pH, or by the interaction with PsbS, that could potentially also affect residue protonations in the vicinity of LutI.

Authors' Reply: we thank the Reviewer for this suggestion. Indeed, a change in protonation state in the vicinity of the LI site could directly alter the CT energies without any other change in the structure.

Our simulation is based on a neutral pH, and our choice of protonation is consistent with the one calculated by Müh et al. on the LHCII trimer. We note that the protonatable sites close to Lut I are only Glu83 and Glu94, which, according to Müh et al., do not change protonation state between pH 6 and pH 8. For this reason, we find improbable that, in the conditions of our MD simulation, a change in lumen protonation would significantly impact the CT energy.

It is important to note that our MD trajectory remains close to the crystal conformation, which should be quenched irrespective of the pH or of the interaction with PsbS. As we have explained in the Conclusions, the first step to understand the effect of transmembrane ΔpH and PsbS should be an unequivocal characterization of the unquenched conformation of LHCII.

(3) Liguori et al. (ref 31) reports that there are large variations at the Chl-a 611-Chl-a 612-LutI cluster of LHCII, even at shorter time scales, compared to the crystal structure. Is this compatible with the small RMSD values for the Chl-a 612/ LutI pair, with respect to the crystal structure conformation, reported in the current study? The Chl-a 612/ LutI and Chl-a 603/ Lut2 sites are located within totally different domains of the LHCII protein scaffold. Chl-a 603/ Lut2 is located close to the monomer-monomer intersections where fluctuations are considerably limited, whereas the Chl-a 612/ LutI is more or less (partly) exposed to the periphery of the LHCII trimer. Thus, the latter could exert more fluctuations. Moreover, a recent study on the LHCII trimer (J. Phys. Chem. B 2019, 123, 45, 9609-9615) reports also fluctuations on the relative helix-A/D orientations in the transition from the light harvesting to the quenched state. Are these fluctuations consistent with what is observed in the current study?

Authors' Reply: Indeed, the simulation by Liguori et al. seems to show more significant fluctuations in the LI site. Compared to their simulation, ours shows generally much lower fluctuations, as commented in the Results section. This could be due to the use of a trimeric structure, but also to the different force field. We note (ref. 32) that the GROMOS force field (used by Liguori et al.) tends to produce “faster” conformational changes than other force fields. This might be one of the reasons why the MD simulations of Liguori et al. explore more of the conformational space than to ours in a comparable simulation time.

We computed the helix A/D torsion parameter as defined in the paper by Daskalakis et al., that is, the ϕ angle of Gly204. We find distributions centered around -70° (Figure S3), and therefore only one of the conformations of Gly204, the “quenched” one, is explored. However, also in the paper by Daskalakis et al. the different monomers never change conformation for G204 (Figure 2B of the paper) in the unbiased MD trajectories. Therefore, our helix A/D torsion fluctuations are compatible with the ones of Daskalakis et al. for unbiased simulations over the considered timescale.

(4) Are all the LHCII monomers in the model sampling the same conformations at the same time? Or is there a heterogeneity within the trimer?

Authors' Reply: The three monomers explore, in general, different parts of the conformational space. This can be seen in the RMSD plots of Figure S1 and is due to the somewhat still limited sampling time. Notably, this effect is consistent with the results of Liguori et al., who find a large variability among the different replicas. However, the rather

low RMSD values in the Lut-Chl dimers suggest that this variability is reduced for our region of interest. In order to assess the effect of this heterogeneity on our results, we used one-way ANOVA to test whether there is significant difference in the CT energies from one monomer to another. For both L1 and L2, this ANOVA test did not allow to prove a significant difference among the monomers ($p > 0.4$).

(5) How far apart are the frames out of the MD trajectory, used for the QM/MM calculations and the subsequent calculation of the mean excited life times? Can we talk about actual lifetimes of the excited state of the LHCII complex, if the input data are separated considerably in time?

Authors' Reply: In order to perform calculations on uncorrelated frames, we extracted frames every 10 ns of trajectory, starting at 1000 ns. The results of the calculations are never taken individually; rather, we only use averaged energies/couplings (note that couplings are averaged as root-medium-square) and energy variances for the reorganization energies. Given the necessity of a compromise between computational cost and statistical efficiency, we chose to have fairly spaced frames to cover most of the simulation time. Our results, thus, correspond to the average lifetime of the excitation in the quenched conformation of LHCII.

(6) The authors claim that the CT state they have characterized within LHCII is able to reduce the LHCII lifetime to less than 300 ps. Can't this be achieved by the EET scheme? The crystal structure of LHCII is highly quenched, which gives high coupling values also between Chl-a 612 Qy/ Lut I S1 states (Biophys. Chem. 2019, 246: 16-24; J. Phys. Chem. B 2019, 123, 45, 9609-9615).

Authors' Reply: We agree with the Reviewer that the EET mechanism is able, depending on the coupling values, to quench Chl excitation and to reduce the LHCII lifetime to even less than 100 ps (Phys. Chem. Chem. Phys. 2015, 17, 15857–15867). Therefore, we do not exclude that the EET mechanism is present in LHCII, neither we claim that the CT mechanism is the only one present in the LHCII, either *in vivo* or in the crystal. Our work is rather focused on showing that the CT mechanism can quench LHCII excitation irrespective of other mechanisms.

Minor issues:

(7) I am not sure I understand the comment on page 3 "To the very best of our knowledge, the electron-transfer mechanism has never been explored to date for real antenna complexes". Do the authors mean the electron transfer between Chl-a/ Lutein within an antenna complex? As such mechanism has been proposed between Chl/ Zeaxanthin in antenna complexes (Science. 320 (2008) 794–797; Science. 307 (2005) 433–436).

Authors' Reply: We thank the Reviewer for pointing out this discrepancy. The remark mentioned here was limited to theoretical atomistic studies, such as the references cited in the previous sentence about the EET mechanism. For the sake of clarity, we modified the sentence as:

“To the very best of our knowledge, the electron-transfer mechanism has never been explored to date by atomistic simulations in real antenna complexes”

(8) Concerning the electrostatic potentials in Fig. 4, how are these calculated? Are these based on one frame (maybe the crystal), or multiple frames? As mentioned also in comment #2 the Chl-a 612/ LutI environment could be highly vulnerable to also protein-protein interactions (LHCII aggregation, or LHCII-PsbS).

Authors' Reply: The electrostatic potential was computed with the APBS program. We added this information in the caption of Figure 4. This potential was computed on one frame among the ones used for the QM calculation. We repeated the potential calculations for several frames along the MD trajectory and found no qualitative differences.

(9) What are the designated residues at the lumen-stroma sides of LHCII in Figure 5? The authors have indicated some residues by solid cylinders, but no reference of their identity is given.

Authors' Reply: We thank the Reviewer for this suggestion. We have added the names/numbers of the residues in Figure 5.

(10) The study by Mennucci's group reports fluctuations of the same state of the LHCII (quenched) over the MD trajectory. In Figure S5 the lifetimes reported cannot be mapped on actual sampled structures of LHCII along the MD trajectory, but these are produced by arbitrarily changing the charge-recombination rate to the ground state. However, the structures in Figure 5 depict LI site in large and small overlap configurations. What would be the change in the couplings, charge-recombination rates & ultimately lifetimes of the LHCII complex for these structures in Figure 5?

Authors' Reply: We thank the Reviewer for the suggestion of directly linking the structures of Figure 5 to the expected lifetime. However, as we demonstrate and analyze in detail in the SI, the CT energy strongly depends on both intermolecular and intramolecular structural variables. Therefore, we cannot directly use the energies of the two structures of Figure 5 to compute a reliable CT energy difference. However, from our linear model we can estimate the change in CT energy following a change in the overlap variable (The coefficients are summarized in Table S4). This gives an estimate of $\sim 2000 \text{ cm}^{-1}$ for the difference in CT energy between the two structures, assuming that the other structural variables do not change, whereas the actual difference in CT energy

between the two structures is $\sim 3300 \text{ cm}^{-1}$. We have added a sentence to the main text explaining this point:

“The overlap values of these structures differ by $\sim 5 \times 10^{-2} \text{ \AA}^3$. Based on the regression coefficients (Supplementary Table S4) we can estimate a $\sim 2000 \text{ cm}^{-1}$ difference in the CT energy, solely due to the change in overlap between these extremes.”

Reviewer #3:

The manuscript by Cupellini et al. represents the results of a very thorough theoretical investigation of the mechanism of chlorophyll excited state dissipation via carotenoids of LHCII complex. The authors derived theoretically for the first time that the charge transfer mechanism is likely to be involved in this process. All previous theoretical studies and some experimental ones were focused upon EET mechanism. Hence this work is of general interest and fundamentally novel contribution to the field of the regulation of light harvesting in the photosynthetic antenna - NPQ. Indeed, only zeaxanthin in the minor LHCII antenna complexes has been proposed as a pigment that could be involved in NPQ. However, the recent research has clearly demonstrated that the quenching in LHCII complex in vivo does not involve zeaxanthin. Moreover, in vitro LHCII devoid of zeaxanthin could undergo large reversible quenching. Therefore, the understanding of the physics and identity of the quencher in LHCII is of great significance. Cupellini et al., not only identifies the quencher and the mechanism of quenching they also explained the reasons behind lutein 1 not lutein 2 being a quencher. This is a very significant insight into the atomic environment of lutein 1 that makes it possible the energy quenching via charge transfer mechanism. Interestingly the transfer times to this carotenoid are consistent with the previously expressed notion that the NPQ quencher is rather slow, hence the economic nature of the regulation of light harvesting in plants, i.e., when the reaction center is actually closed, then NPQ starts to work, but when it is open, it gains priority for the captured energy.

Authors' Reply: we sincerely thank the Reviewer for the very positive comments.

Reviewer #1 (Remarks to the Author):

The authors have addressed my questions and requests, which include:

1) Adding the histogram of excitation energies. What I see is indeed a normal distribution, therefore the student's t test is appropriate.

2) Authors have included the QM/MM calculation on the X-ray structure.

I recommend publication in Nature Chemistry.

Reviewer #2 (Remarks to the Author):

The authors have provided convincing details based on the comments of the reviewers. There are certainly major approximations involved in the study. However, the calculations are nicely performed and they can provide at least an indication that the CT mechanism could contribute partially to the quenched state of the LHCII trimer. I therefore believe that the conclusions are indeed original, and they will influence thinking in the field. The calculations can be reproduced given the details provided.